# Acute Toxicity Evaluation of Phosphatidylcholine Nanoliposomes Containing Nisin in *Caenorhabditis elegans*

**DOI:** 10.3390/molecules28020563

**Published:** 2023-01-05

**Authors:** Juliana Ferreira Boelter, Solange Cristina Garcia, Gabriela Göethel, Mariele Feiffer Charão, Livia Marchi de Melo, Adriano Brandelli

**Affiliations:** 1Laboratory of Biochemistry and Applied Microbiology, Institute of Food Science and Technology, Federal University of Rio Grande do Sul, Porto Alegre 91501-970, Brazil; 2Laboratory of Toxicology, Faculty of Pharmacy, Federal University of Rio Grande do Sul, Porto Alegre 90610-000, Brazil; 3Laboratory of Toxicological Analyses, Institute of Health Sciences, Feevale University, Novo Hamburgo 93525-075, Brazil

**Keywords:** nanoencapsulation, nanotoxicology, antimicrobial peptide, liposomes, food preservative

## Abstract

Liposomes are among the most studied nanostructures. They are effective carriers of active substances both in the clinical field, such as delivering genes and drugs, and in the food industry, such as promoting the controlled release of bioactive substances, including food preservatives. However, toxicological screenings must be performed to ensure the safety of nanoformulations. In this study, the nematode *Caenorhabditis elegans* was used as an alternative model to investigate the potential in vivo toxicity of nanoliposomes encapsulating the antimicrobial peptide nisin. The effects of liposomes containing nisin, control liposomes, and free nisin were evaluated through the survival rate, lethal dose (LD_50_), nematode development rate, and oxidative stress status by performing mutant strain, TBARS, and ROS analyses. Due to its low toxicity, it was not possible to experimentally determine the LD_50_ of liposomes. The survival rates of control liposomes and nisin-loaded liposomes were 94.3 and 73.6%, respectively. The LD_50_ of free nisin was calculated as 0.239 mg mL^−1^. Free nisin at a concentration of 0.2 mg mL^−1^ significantly affected the development of *C. elegans*, which was 25% smaller than the control and liposome-treated samples. A significant increase in ROS levels was observed after exposure to the highest concentrations of liposomes and free nisin, coinciding with a significant increase in catalase levels. The treatments induced lipid peroxidation as evaluated by TBARS assay. Liposome encapsulation reduces the deleterious effect on *C. elegans* and can be considered a nontoxic delivery system for nisin.

## 1. Introduction

Nanotechnology has emerged as a new and promising science, revolutionizing the production process of many materials from electronics to those used in the pharmaceutical and food industries [1,2]. However, nanoscale materials have different properties compared to the same material on bulk state, and it is not possible to predict their biological activity by extrapolating the physical and chemical properties of the components that constitute these nanostructured materials [3,4]. 

Nanostructures based on biodegradable materials, such as natural and synthetic polymers and lipids, hold enormous potential as effective drug delivery systems [5]. Liposomes are one of the most commonly applied organic nanostructures as they can be produced using natural ingredients scaled to an industrial level and can encapsulate both hydrophobic and hydrophilic compounds. Liposomes can be defined as colloidal structures formed by a suitable combination of constituent molecules in an aqueous solution [6]. When amphiphilic molecules, such as phospholipids, are placed in an aqueous environment, they form complex aggregates to protect their hydrophobic sections from water molecules, maintaining contact with the aqueous phase by hydrophilic groups. If sufficient energy is provided to the phospholipid aggregates, they can arrange themselves in an orderly manner in a closed vesicular bilayer [7]. Encapsulation into nanoliposomes promotes an improvement in the properties of bioactive compounds because it protects the compounds from degradation and interactions with undesirable compounds and increases their efficiency and apparent solubility [6,8].

Despite growing research on nanostructured food antimicrobials [9], there is still little information about the toxicity of nanostructures intended for food application. These nanomaterials added to food are ingested and pass through the digestive system. Nanostructures incorporated into the food package may be released to the food matrix and reach the gastrointestinal tract as well [10]. Because a novel formulation can be beneficial or deleterious to an organism, it is important to understand its behavior and mechanism of action in biological systems in order to establish the potential risks to consumers [11,12]. 

A new food or medicine must be safe to be approved for industrialization. The safety assessment of nanomaterials to human health and the environment is one of the greatest challenges in the field of nanotoxicology [12]. The large number of novel nanomaterials requires the use of in vivo models to provide adequate toxicological screening. However, the traditional models used in toxicological studies, especially those with mammals, have been the subject of many discussions because of the number of individuals required and the suffering caused during some kinds of experiments [13]. In addition, maintenance of these laboratory animals involves elevated costs. Thus, alternative methods and models have emerged from the scientific community to reassess the use of mammalian experiments [12,14]. 

In this study, we employed the free-living nematode *Caenorhabditis elegans*, which has been shown to be a useful model for testing many chemicals, including metals [15,16], persistent organic pollutants [17], pesticides [18], and nanomaterials [19,20]. *C elegans* is a small and transparent nematode that lives mainly in the liquid phase of soil and feeds on soil microorganisms. These organisms have a short life cycle, and their laboratory maintenance is easy and inexpensive [21]. *C. elegans* was the first multicellular organism to have its genome completely sequenced, and it is one of the best characterized animals at the genetic, physiological, molecular, and developmental levels [22]. Moreover, its genome shows about 70% of homology with humans [23]. Currently, knockout strains for genes of interest and transgenic nematodes expressing green fluorescent protein (GFP)-tagged proteins are available, making *C. elegans* an ideal model for expression or protein localization studies [23,24]. 

The objective of this study was to evaluate the acute toxicity of phosphatidylcholine liposomes encapsulating the antimicrobial peptide nisin using *C. elegans* as an alternative in vivo model. Additionally, the potential involvement of oxidative stress mechanisms was investigated. 

## 2. Results

### 2.1. Liposome Encapsulation of Nisin

Liposomes were prepared by the thin-film hydration method using a 1:15 (w/w) nisin to phosphatidylcholine ratio. Unloaded liposomes had a mean diameter of 125 ± 6 nm, polydispersity index (PDI) of 0.206 ± 0.08, and zeta potential +3.43 ± 2.96 mV. Liposomes loaded with nisin had mean diameter of 168 ± 27 nm, PDI of 0.267 ± 0.11, and zeta potential of -+5.44 ± 2.27 mV. The entrapment efficiency was 100%. These values were in the expected range for nisin encapsulated into liposomes prepared with phosphatidylcholine [25].

### 2.2. Dose–Response Curves for Liposomes and Free Nisin

The survival of nematodes was evaluated after exposure to different concentrations of free nisin and liposomes. Dose–response curves were constructed, and the harmful effect of free nisin could be observed for concentrations higher than 0.2 mg mL^−1^ (Figure 1). The LD_50_ for free nisin was determined as 0.239 mg mL^−1^. At the highest concentration tested, the survival rate of *C. elegans* was 14.2% for free nisin and 73.6% for nisin-loaded liposomes. Empty liposomes caused no significant reduction of the survival rate (94.3%). Thus, the determination of LD_50_ for liposomes loaded with nisin and control liposomes was not experimentally possible. The survival rate for the vehicle control (0.085 mol L^−1^ NaCl) was 100%.

### 2.3. Development of Nematodes

The size of the nematodes was evaluated, and the values for *C. elegans* treated with 0.1 mg mL^−1^ nisin-loaded liposomes or control empty liposomes were significantly higher than the control, which showed a mean surface area of 0.104 mm^2^. The remaining concentrations of liposomes were not significantly different from the control or the lowest concentration of free nisin. However, the exposure to free nisin concentrations from 0.2 mg mL^−1^ led to a decrease of approximately 25% of body area (Figure 2), with a mean surface area of 0.075 mm^2^. This result suggested that the normal development of *C. elegans* was affected by nisin exposure.

### 2.4. ROS Levels

ROS levels were determined using 2′7′-dichlorofluorescein-diacetate (H_2_DCF-DA) dye. The exposure of *C. elegans* to liposomes and free nisin caused a significant increase in H_2_DCF-DA oxidation, reflecting the generation of ROS (Table 1). The maximum tested concentration of control and nisin liposomes generated higher ROS levels than the highest concentration of free nisin. Only the lowest concentration of free nisin generated less ROS than the control with saline solution, which was set as 100% fluorescence (Table 1).

### 2.5. Fluorescence Quantification of Antioxidant Enzymes

The green fluorescent protein (GFP)-expressing strains CF1553 (muls84) for superoxide dismutase (SOD) and GA800 (wuls154) for catalase (CAT) were exposed to nisin and liposomes, and the total GFP fluorescence response to each treatment was determined. A significant increase in CAT levels (strain GA800) was observed following treatments with liposomes containing nisin and free nisin (Figure 3A) at higher concentrations but not with other treatments. However, the same treatments induced an increase in SOD levels only at the lower concentration (Figure 3B), indicating a random effect on expression of SOD-3. Control liposomes did not induce an increase in CAT or SOD levels, suggesting that increased levels of antioxidant enzymes were induced under specific conditions.

Illustrative images of control and nisin-treated *C. elegans* for GFP-CAT (Figure 3C,D) and GFP-SOD (Figure 3E,F) are displayed, showing that CAT and SOD were overexpressed in comparison to controls.

### 2.6. Lipid Peroxidation

Thiobarbituric acid reactive substances (TBARS) are formed as by-products of lipid peroxidation and are a signal of oxidative damage. The effect of control liposomes, nisin-loaded liposomes, and free nisin on generation of TBARS was evaluated. The TBARS value for saline solution used as control was 26.4 μg MDA g^−1^ protein, increasing to values ranging from 38.2 to 50.9 μg MDA g^−1^ protein for 0.1 mg mL^−1^ nisin liposomes and 0.2 mg mL^−1^ free nisin, respectively. There were significant differences between all treatments compared to the control (Figure 4). However, no differences in TBARS levels were observed among treatments. The induction of ROS was reflected in lipid peroxidation.

### 2.7. Correlation of Data

The main data of this study were subjected to Spearman’s correlation, and the significant ones are listed in Table 2. The test revealed a positive significant correlation between death rate (obtained from the survival experiment) and the mean quantity of ROS for all treatments. In liposomes containing nisin, the death rate was inversely correlated to the size of nematodes, i.e., the larger the organism, the lower the death rate.

## 3. Discussion

Liposomes have been gaining importance as nanostructures due to the countless possibilities of lipid combinations, substances to be encapsulated, and preparation methods. These nanostructures can be interesting mediums to deliver bioactive peptides [5]. In vivo assays are very relevant as they offer information about integrated biological effects of nanoparticles. However, some studies have indicated that MTT cell viability assay, one of the most used tests for cytotoxicity, is not compatible with the lipid nature of liposomes and lead to incorrect conclusions [26]. Thus, in accordance with the 3Rs policy, *C. elegans* represents an alternative method for toxicological studies of liposome formulations. Results obtained using the *C. elegans* model can be crucial in establishing new approaches in nanotoxicology and in predicting their effects in complex animal models [19].

In this work, a significant reduction of the survival rate of *C. elegans* was observed with exposure to nisin concentrations higher than 0.2 mg mL^−1^. Limited information is available about the response of *C. elegans* to exogenous antimicrobial peptides. Sublethal concentrations of temporin-1Tb and esculentin-derived peptides increased the survival of bacterial-infected *C. elegans* after 40 h treatment. In contrast, 64 μM of temporin-1Tl killed *C. elegans*, a dose that caused in vitro toxicity to erythrocytes as well [27]. Moreover, the antimicrobial peptide brevinin-2ISb increased the survival rate of MRSA-infected *C. elegans* by inducing the expression of innate immune genes [28]. 

Nisin is used as a food preservative in more than 50 countries. More recently, in addition to the food industry, nisin has also been studied for use in the medical field to control common oral diseases, to combat animal infections, and even as an antitumor drug [29]. Several studies have pointed out that nisin has no chronic or subchronic toxic effect, reproductive toxicity, genotoxicity, and carcinogenic or teratogenic effects [30,31]. However, a certain degree of toxicity or adverse effects have been reported at high concentrations of nisin [31,32], and toxicity studies on nanoformulations containing nisin are very scarce. In the present study, no significant toxicity was observed for control or nisin-loaded PC liposomes, with the highest tested concentrations killing 5.7% and 26.4% of *C. elegans* population, respectively. PC liposomes have been considered safe for several uses [33,34]. In fact, it has been reported that exposure to liposomes caused no adverse effects in the life span of *C. elegans* [35] and that hybrid PC liposomes with carbon nanotubes did not affect the fertility of the nematodes [36].

In this study, it was observed that encapsulation protected against the possible deleterious effects of nisin. Free nisin at the concentration of 0.25 mg mL^−1^ killed 54% of exposed nematodes, while 100% of *C. elegans* exposed to the same concentration of encapsulated nisin survived. This agrees with some reports that have indicated that nanoencapsulation decreases the extent and the types of nonspecific toxicities and allows an increase in the amount of drug that can be effectively delivered [37,38]. However, it is important to point out that harmful effects on *C. elegans* were observed at higher concentrations than those usually employed in food preservation (12.5 mg kg^−1^) and the acceptable daily intake of 0.13 mg kg^−1^ body weight [39].

The growth of *C. elegans* is determined by a conservative genetic pathway and is considered a good parameter to evaluate toxic effects in this model [40]. The lowest concentrations of PC liposomes tested in this work promoted enlargement of the nematodes. In this regard, an increase in the lifespan of *C. elegans* was previously observed by treatment with 0.1 mg mL^−1^ PC [41], the same concentration that caused increased body size in the present study. Some authors have shown that subjecting *C. elegans* to mild degrees of stress can induce thermotolerance, longer lifespan, and greater resistance to oxidative stress in a process known as hormesis [42,43]. We suggest that this increase in the nematode size may be due to hormesis. In contrast, higher concentrations of free nisin affected the development of *C. elegans*, which was about 25% smaller than the control. A similar rate of size reduction was observed in *C. elegans* exposed to 50 ppm of silver nanoparticles, fumed SiO_2_, and multiwalled carbon nanotubes [44]. In nematodes exposed to liposomes containing nisin, as the death rate increased, their size decreased. These results indicate that encapsulation protects against the potential deleterious effect of free nisin on *C. elegans*.

Oxidative stress occurs when there is an imbalance between the generation of ROS and the antioxidant defenses of the organism. Oxidative stress is associated with the toxicity mechanism of many compounds as it may cause damage to proteins, lipids, and DNA [45]. Here, we observed a rise in ROS production in *C. elegans* treated with liposomes and free nisin, reaching up to a five-fold increase compared to control samples. However, this value was reached with 0.25 mg mL^−1^ of free nisin and 0.8 mg mL^−1^ of nanoencapsulated nisin, suggesting that encapsulation reduces the ROS-inducing effects. Spearman’s correlation test showed a positive correlation between death rate and ROS levels in all treatments. Other factors may be contributing to the relatively high death rate of free nisin. Osmotic stress could be related to an increase in ROS as oxidative stress is the conserved mechanism of adaptive response, and mild oxidative stress can increase resistance when organisms are challenged with higher doses of a particular stressor [46]. In this regard, Nisaplin^®^ formulation has NaCl as solid fillers, and previous tests have shown the death of 80% of the nematode population exposed to 9 mg mL^−1^ saline solution (Boelter et al., 2023, in preparation). The increase in ROS levels at the higher concentrations of control liposomes suggests that the nanocapsule itself can induce oxidative stress. This effect could be associated with the high surface area of nanoparticles, which leads to an increase in chemical reactivity and results in higher specific interactions with biological structures, thus increasing oxidation [47,48]. However, it is important to emphasize that high doses are not usual in nanotoxicology as the nanostructures are often more efficient than the drug in its free form.

Antioxidant enzymes, such as superoxide dismutase (SOD), glutathione peroxidase (GPX), and catalase (CAT), are important in the preservation of homeostasis for normal cell function and are frequently used as markers of oxidative stress. CAT levels were significantly increased at the highest concentration of liposomes containing nisin and free nisin, where death rates were also higher. However, increased SOD levels were detected at lowest concentration of free nisin and high dose of nisin liposomes. *C. elegans* possibly contains one of the largest contingents of SOD proteins among aerobic organisms, including five *sod* genes established by homology analysis. However, a mutational analysis provided evidence that *C. elegans* carrying deletions of all *sod* genes showed no tissue damage and survived as long as wild-type control nematodes [49]. In this work, nisin apparently caused a random effect on the GF-mutant CF1553 for SOD-3, suggesting that the effect of nisin on *C. elegans* could be associated with the expression of other SOD isoforms. Up-regulation of antioxidant enzymes may be a compensatory effect to combat oxidative stress [50], although increased ROS levels does not always involve increased antioxidant defenses and cell death. In *C. elegans*, SOD and CAT are part of a broad spectrum of antioxidant mechanisms that contribute to protection against oxidative damage. These comprise a large array of antioxidant or drug-metabolizing enzymes that affect phase 2 detoxification, including glutathione-S-transferases (GST) and UDP-glucosyltransferases. For example, *C. elegans* has more isoforms of CAT and SOD than higher animals [51], and *C. elegans* survival during starvation is partially promoted by upregulation of antioxidant enzymes, such as GST-4 and GST-10 [52]. 

The increase in TBARS values in all treatments in comparison to control suggested that exposure of *C. elegans* to free or encapsulated nisin induced lipid peroxidation. However, this effect was not correlated with the death rate, indicating that other oxidative damage could be induced by ROS instead of lethal injury to membranes. In contrast, the antioxidant effect of melatonin was enhanced by encapsulation into lipid-core nanocapsules, reducing TBARS values in *C. elegans* compared to free melatonin [38].

The antimicrobial effect of nisin against Gram-positive bacteria could be achieved at 0.012 mg mL^−1^ [53], and significant effects on survival, development, and ROS induction in *C. elegans* were observed at doses higher than 0.1 mg mL^−1^ for free nisin or even higher concentrations for encapsulated nisin. The present study suggests that liposomes represent a nontoxic delivery system for nisin, confirming these nanostructures as a good alternative for delivery of this antimicrobial peptide in food.

## 4. Materials and Methods

### 4.1. Materials

Phospholipon^®^ 90G (pure phosphatidylcholine stabilized with 0.1% ascorbyl palmitate) was supplied by Lipoid GMBH (Ludwigshafen, Germany). Commercial nisin (Nisaplin^®^) was purchased from Danisco Brasil (Cotia, Brazil). According to the manufacturer, the formulation contains NaCl and denatured milk solids as fillers and 2.5% pure nisin. Nisin was dissolved in 0.1 mol/L HCl and then diluted with a phosphate buffer of pH 7.0. Bacto agar and bacto peptone were obtained from Becton Dickinson BD^®^ (Franklin Lakes, NJ, USA) and HiMedia Laboratories^®^ (Mumbai, India), respectively. 1,1,3,3-Tetramethoxypropane and 2′,7′-dichlorofluorescein-diacetate (H_2_DCF-DA) were supplied by Sigma-Aldrich (St. Louis, MO, USA). Phosphoric acid and 2-thiobarbituric acid were purchased from Tedia Co (Fairfield, OH, USA) and Spectrum Chemical Co (Gardena, CA, USA), respectively.

### 4.2. Liposome Preparation and Characterization

Liposomes were produced by the thin-film hydration method [54]. Briefly, 76 mg of Phospholipon^®^ 90G was dissolved in 15 mL chloroform in a round-bottom flask, and the organic solvent was removed using a rotary evaporator until a thin film was formed on the walls. After 24 h in a desiccator, 5 mL of 1 mg mL^−1^ nisin solution was added to disperse the resulting dried lipid film. These mixtures were homogenized at 60 °C and sonicated in an ultrasonic cell disruptor (Unique, Brazil) by five cycles of 1 min at intervals of 1 min, during which the samples were kept in ice. Then, the solution was filtered through 0.22 µm membranes. The drug-unloaded liposomes are denominated as control liposomes, and the volumes used in all tests were the same as nisin-loaded liposomes. Liposomes were characterized as described elsewhere [55]. Particle size, polydispersity index (PDI), and zeta potential (ζ) were determined after dilution in ultrapure water using a Zetasizernano-ZS ZEN 3600 equipment (Malvern Instruments, Herrenberg, Germany).

### 4.3. Strains, Culture, and Synchronization of C. elegans

The *C. elegans* strains N2 bristol (wild type), CF1553 (muls84), and GA800 (wuls151) were provided by the *Caenorhabditis* Genetics Center (University of Minnesota, Twin Cities, MN, USA) and maintained and handled at 20 °C on *Escherichia coli* OP50 in NGM (nematode growth medium) plates. Synchronous L1 population was obtained by isolating embryos from gravid hermaphrodites using bleaching solution (1% NaOCl, 0.25 mol L^−1^ NaOH), followed by floatation on a sucrose gradient to segregate eggs from dissolved worms and bacterial debris in accordance with standard procedures [56]. Eggs were washed with M9 buffer (0.02 mol L^−1^ KH_2_PO_4_, 0.04 mol L^−1^ Na_2_HPO_4_, 0.08 mol L^−1^ NaCl, and 0.001 mol L^−1^ MgSO_4_) and allowed to hatch in unseeded NGM overnight.

### 4.4. Exposure to Liposomes with Nisin, Control Liposomes, and Free Nisin

Synchronized L1 worms were used (2500 nematodes for LD_50_ determination, thiobarbituric acid reactive substances evaluation, and development assays and 1500 nematodes for measurement of reactive oxygen species and fluorescence quantification of CAT and SOD). The nematodes were immersed in 0.085 mol L^−1^ NaCl solution and exposed for 30 min to liposomes containing nisin, control liposomes, and free nisin. The control samples contained only the nematodes and NaCl solution. Free nisin was tested from 0.05 to 0.3 mg mL^−1^, while nisin concentration in liposomes ranged from 0.05 to 0.8 mg mL^−1^ (corresponding to an amount of PC ranging from 0.75 to 12 mg mL^−1^). Similar amounts of empty liposomes were used as control. After exposure, nematodes were washed 3 times with saline solution (0.085 mol L^−1^ NaCl).

### 4.5. LD_50_ Determination

To determine the LD_50_ of liposomes and free nisin, the nematodes were exposed and washed as outlined in Section 4.4 and then placed on OP50-seeded NGM plates. The number of surviving worms on each plate was determined 24 h after exposure. All of the tested doses were compared to the control group, which did not receive treatment. Three replicates were performed. The dose–response curves were drawn according to a sigmoidal model with a top constraint at 100%.

### 4.6. Development of Nematodes

For the evaluation of *C. elegans* development, the surface area of 20 adult nematodes per treatment was measured. For this, the NGM plates were washed with distilled water and the nematodes were transferred to centrifuge tubes, being washed successively until the solution was clear. After this procedure, 15 µL of the solution with the worms were mounted on 2% agarose pads with 15 µL of levamisole (22.5 mg mL^−1^) to anesthetize. Pictures were taken, and the flat surface area of nematodes was measured using the AxioVision software LE (version 4.8.2.0 for windows). Results were expressed as percentage of body area relative to control, which was taken as 100% [19].

### 4.7. Measurement of Reactive Oxygen Species (ROS)

After the exposure, nematodes were maintained in 100 µL of saline solution (0.085 mol L^−1^ NaCl) and transferred to a 96-well plate, and 2′7′-dichlorofluorescein-diacetate (H_2_DCF-DA) was added to reach a final concentration of 0.05 mmol L^−1^. The fluorescence levels were measured at an excitation λ = 485 nm and emission λ = 535 nm using a microplate reader (Spectramax Me2; Molecular DevicesLLC, Sunnyvale, CA, USA) at 20°C. The fluorescence from each well was measured for 90 min at 10 min intervals. Results were expressed as percentage of fluorescence intensity relative to control wells, which were taken as 100%.

### 4.8. Fluorescence Quantification

The green fluorescent protein (GFP)-expressing strains CF1553 (muls84) for superoxide dismutase (SOD) and GA800 (wuls154) for catalase (CAT) were submitted to acute exposure as described above. Nematodes were maintained in 100 µL saline solution and transferred to a 96-well plate, where total GFP fluorescence was measured using 485 nm excitation and 530 nm emission filters using a microplate reader (Spectramax Me2; Molecular DevicesLLC, Sunnyvale, CA, USA) at 20 °C. The fluorescence from each well was measured for 10 min at 1 min intervals. Results were expressed as the mean percentage of fluorescence intensity relative to control wells, which were taken as 100%.

### 4.9. Fluorescence Microscopy

For each treatment, a slide was taken and 20 nematodes were mounted on 2% agarose pads and anesthetized with 15 µL of levamisole (22.5 mg mL^−1^). Fluorescence observations were performed for image acquisitions using an Olympus IX-71 fluorescence microscope (Olympus, Tokyo, Japan).

### 4.10. Lipid Peroxidation

Thiobarbituric acid reactive substances (TBARS) were determined as a marker of lipid peroxidation in adult nematodes. TBA (thiobarbituric acid) assay using a 1,1,3,3-tetramethoxypropane solution as malondialdehyde (MDA) standard was employed [38]. After 48 h exposure, the plates containing the nematodes were washed to remove the OP50 medium. The nematodes were disrupted in Turrax homogenizer at full amplitude for 1 min to release the lipid and protein content. After centrifugation at 10,000× *g* for 5 min, the supernatant was transferred to cryotubes, where the TBARS reaction took place, with the addition of 0.1 mol L^−1^ phosphoric acid solution, 0.02 mol L^−1^ sodium dodecyl sulfate solution, and 0.04 mol L^−1^ 2-thiobarbituric acid solution. The reaction was developed for 90 min at 100 °C under agitation in a water bath. After stopping in an ice bath, the samples were transferred to 96-well plates, and the absorbance was read at 532 nm (Spectramax Me2; Molecular DevicesLLC, Sunnyvale, CA, USA). The protein content of the samples was determined by Coomassie blue binding assay [57].

### 4.11. Statistical Analysis 

The results were subjected to variance analysis (ANOVA), and means were compared by the Tukey test at a level of 5% significance. The results obtained with the methods outlined in Section 4.4, Section 4.5 and Section 4.6 were submitted to Spearman’s correlation at a level of 5% significance. All tests were performed using the Prism 5.0 software (GraphPad Software Inc., La Jolla, CA, USA).

## Figures and Tables

**Figure 1 molecules-28-00563-f001:**
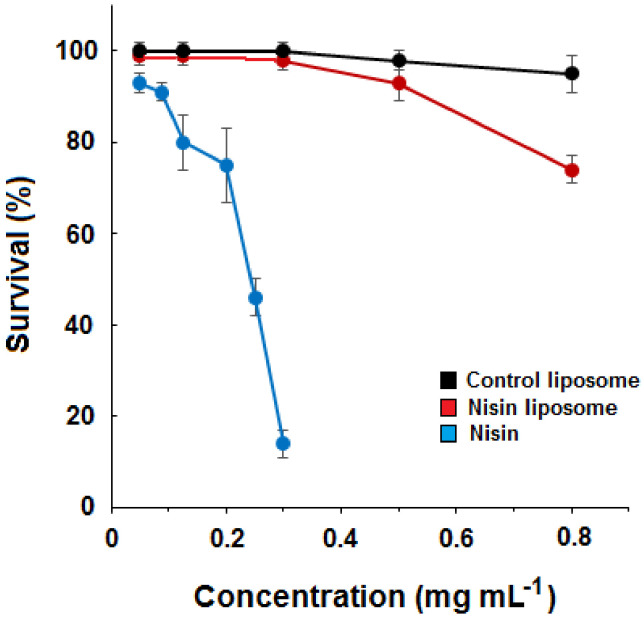
Survival of *C. elegans* exposed to different concentrations of nisin and nisin-loaded liposomes. Nematodes were incubated with increasing concentrations of free nisin (blue symbols) or liposomes containing the equivalent amount of encapsulated nisin (red symbols). Empty liposomes (black symbols) were used as control. Vehicle control was performed with saline solution (0.085 mol L^−1^ NaCl). The number of surviving nematodes was determined 24 h after exposure. Values are the means ± standard deviations of three independent experiments.

**Figure 2 molecules-28-00563-f002:**
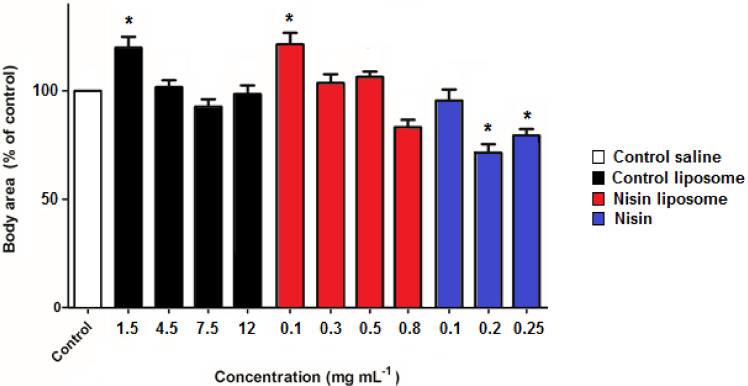
Influence of nisin and liposomes on the development of *C. elegans*. Controls were performed by incubation in saline solution (0.085 mol L^−1^ NaCl) or in the presence of empty liposomes. Values for control liposomes are expressed as PC concentration, corresponding to the equivalent PC amount of liposomes containing 0.1, 0.3, 0.5, and 0.8 mg mL^−1^ nisin, respectively. Values are the means ± standard deviations of three independent experiments. * denotes significant differences with control saline (*p* < 0.05).

**Figure 3 molecules-28-00563-f003:**
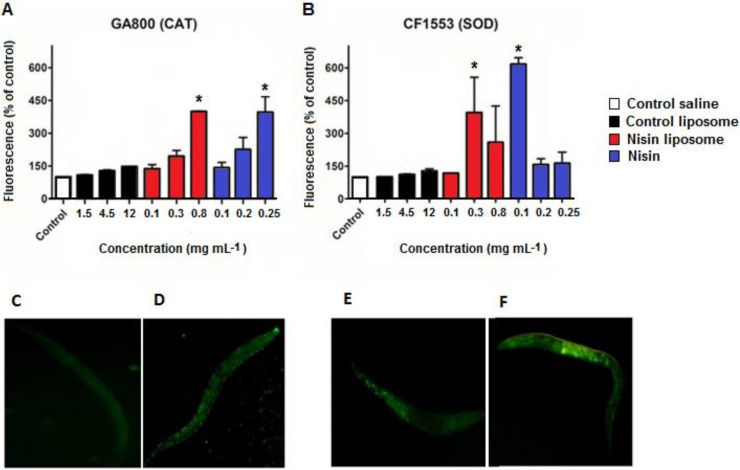
Quantification of fluorescence of (**A**) GA800 (CAT) and (**B**) CF1553 (SOD) strains after exposure to liposomes with nisin (red) or free nisin (blue). Controls were performed by incubation in saline solution (0.085 mol L^−1^ NaCl) or in the presence of empty liposomes. Values for control liposomes are expressed as PC concentration, corresponding to the equivalent PC amount of liposomes containing 0.1, 0.3, and 0.8 mg mL^−1^ nisin, respectively. The photographs are representative of the control animals (**C**,**E**) for CAT and SOD, respectively, and of the samples with higher fluorescence: CAT for nisin at 0.25 mg mL^−1^ (**D**) and SOD for nisin at 0.1 mg mL^−1^ (**F**). Values are the means ± standard deviations of three independent experiments. * denotes significant differences with control saline (*p* < 0.05), which was set as 100%.

**Figure 4 molecules-28-00563-f004:**
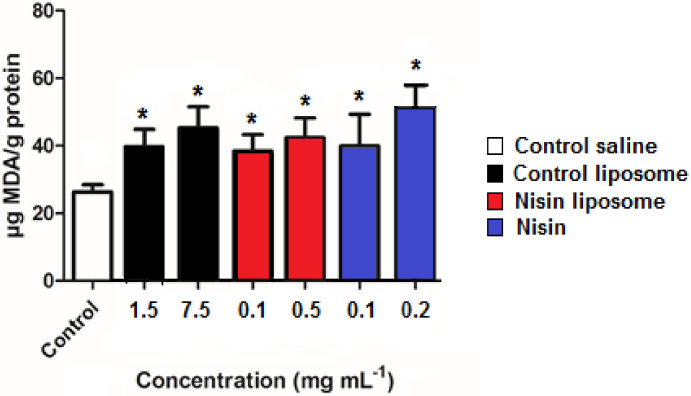
Quantification of thiobarbituric acid reactive substances in adult nematodes after exposure to free nisin (blue) or liposomes containing nisin (red). Controls were performed by incubation in saline solution (0.085 mol L^−1^ NaCl) or in the presence of empty liposomes. Values for control liposomes are expressed as PC concentration, corresponding to the equivalent PC amount of liposomes containing 0.1 and 0.5 mg mL^−1^ nisin, respectively. Values are the means ± standard deviations of three independent experiments. * denotes significant differences with control saline (*p* < 0.05).

**Table 1 molecules-28-00563-t001:** Reactive oxygen species levels measured by fluorescence emission from 2′7′-dichlorofluorescein-diacetate dye.

Sample	Dose (mg mL^−1^)	Relative Fluorescence (%) ^1^
Free nisin	0.1	61 ± 10 *
	0.2	161 ± 25 *
	0.25	321 ± 191 *
Nisin liposome	0.1	136 ± 76
	0.3	131 ± 75
	0.5	144 ± 65
	0.8	514 ± 251 *
Control liposome ^2^	1.5	176 ± 73
	4.5	162 ± 56
	7.5	338 ± 250 *
	12	577 ± 203 *

^1^ The fluorescence of vehicle control (0.085 mol L^−1^ NaCl) was set as 100%. Values are the means ± standard deviations of three independent experiments. * denotes significant differences with control saline (*p* < 0.05). ^2^ Values are expressed as PC concentration for control liposomes, corresponding to the equivalent PC amount of liposomes containing 0.1, 0.3, 0.5, and 0.8 mg mL^−1^ nisin, respectively.

**Table 2 molecules-28-00563-t002:** Spearman’s correlation test of death rate, reactive oxygen species (ROS), and size of animals exposed to free nisin, liposomes containing nisin, and control liposomes.

Treatment	Correlation	*r*	*p* Value
Free nisin	Death rate x ROS	0.7667	0.0214
Nisin-loaded liposome	Death rate x ROS	0.7486	0.0255
	Death rate x Size	−0.8398	0.0061
Control liposome	Death rate x ROS	0.7886	0.0172

## Data Availability

The data presented in this study are available on request from the corresponding author.

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
