# Peer review of "Acute Toxicity Evaluation of Phosphatidylcholine Nanoliposomes Containing Nisin in Caenorhabditis elegans"

_molecules, 2023, doi:10.3390/molecules28020563_

Round 1

Reviewer 1 Report

The authors bring an excellent study design with a significant problem to investigate. Although, the analysis performed is not adequate for the Molecules script. I recommend sending it to another journal. I wish you luck.

Author Response

Authors thank the Reviewer for the time and effort to evaluate the manuscript and for recognizing the merits of the study. The article was improved in accordance to the pertinent remarks, and we expect the revised manuscript fulfill the requirements to be published in Molecules.

Reviewer 2 Report

In this manuscript, the authors used C. elegans as a model organism to examine the potential in vivo toxicity of nanoliposomes encapsulating the antimicrobial peptide nisin. The results showed that liposome encapsulation reduced the deleterious effect of nisin on C. elegans, suggesting liposome can be considered a non-toxic delivery system for nisin in vivo. It is an interesting study. However, the authors should address the following reviewer’s concerns before the manuscript is accepted for publication.

1. Figure 1

The scale for the X-axis is not clear because the ticks intervals are not consistent. The authors should describe it either in the main text or in the figure legend.

2. Figure 2

How the control worms were treated is not descried in the manuscript. Were the control worms treated by empty liposomes or without any treatment?

3. Figure 3

·      Similar to Figure 2, the control worms are not defined.

·      The expression of CAT is increased when the concentrations of nisin and nisin liposomes are increased, which is consistent with their effect on the ROS levels. However, the SOD expression data is confusing. The strain CF1553 is used to examine the expression of SOD-3. The current data suggest the effect of nisin and  nisin liposomes on the expression of SOD-3 is a random effect. C. elegans has five SOD isoforms. The authors should check the expression of other SODs (SOD-1, SOD-2, SOD-4 and SOD-5) using either GFP marker or qPCR. It is possible Nisin actually influences the expression of some of these SODs.

Author Response

In this manuscript, the authors used C. elegans as a model organism to examine the potential in vivo toxicity of nanoliposomes encapsulating the antimicrobial peptide nisin. The results showed that liposome encapsulation reduced the deleterious effect of nisin on C. elegans, suggesting liposome can be considered a non-toxic delivery system for nisin in vivo. It is an interesting study. However, the authors should address the following reviewer’s concerns before the manuscript is accepted for publication.

Answer: Authors thank the Reviewer for the time and effort to evaluate the manuscript and for recognizing the merits of the study. The manuscript was improved in accordance to the pertinent remarks.

  1. Figure 1

The scale for the X-axis is not clear because the ticks intervals are not consistent. The authors should describe it either in the main text or in the figure legend.

Answer: The X-axis of Figure 1 was originally plotted as log scale; it revised to a linear scale to make the intervals clear.

  1. Figure 2

How the control worms were treated is not descried in the manuscript. Were the control worms treated by empty liposomes or without any treatment?

Answer: Controls were performed by treatments with empty liposomes and saline solution. This information was included in Figure 2 and Methods section of the revised manuscript (lines 131-134 and 360-361).

  1. Figure 3

- Similar to Figure 2, the control worms are not defined.

Answer: Controls were performed by treatments with empty liposomes and saline solution. This information was included in Figure 3 (lines 173-176).

- The expression of CAT is increased when the concentrations of nisin and nisin liposomes are increased, which is consistent with their effect on the ROS levels. However, the SOD expression data is confusing. The strain CF1553 is used to examine the expression of SOD-3. The current data suggest the effect of nisin and nisin liposomes on the expression of SOD-3 is a random effect. C. elegans has five SOD isoforms. The authors should check the expression of other SODs (SOD-1, SOD-2, SOD-4 and SOD-5) using either GFP marker or qPCR. It is possible Nisin actually influences the expression of some of these SODs.

Answer: Authors thank for the relevant remark. In this regard, the discussion was improved to indicate the random effect of the treatments on SOD-3 expression and about the need to check for other SOD isoforms (lines 161-163 and 288-294 in the revised manuscript). The suggestion to check expression of other SODs by GFP marker or qPCR will be incorporated in future studies, but unfortunately we have no access to additional mutant GFP strains at this time. Moreover, the inclusion of a robust qPCR protocol will require reasonable time for set up and standardization. 

Reviewer 3 Report

In this study, the authors used C. elegans as a model to investigate the in vivo toxicity of an antimicrobial peptide nisin encapsulated in nanoliposomes. Here, using physiological and biochemical experiments, the authors provide evidence for liposome encapsulation-mediated reduced toxicity of nisin in C. elegans.

Although the authors have performed the required experiments, they still need to include appropriate controls. Also, the methodology, experimental result description, and discussion must be improved. 

The incorporation of the following suggestions will improve the result interpretations.

  1. Dose-response curve - Vehicle control is required. Also, it should be clearly stated at what time points were chosen and why. A survival plot of C. elegans treated with nisin (0.239 mg mL-1 ) along with appropriate controls will be helpful.
  2. Development of nematodes - It needs to be clarified from the graph what control was used. It is counter-intuitive that the worms treated with Nisin+liposome 0.1 mg/mL-1 have larger body sizes, and the Authors should comment on it. Typically, developmental assay in C. elegans includes tracking the development of worms from the egg to the adult stage. This can be done by using a molting GFP marker or visually assessing the developmental stage. 
  3. ROS levels - must include vehicle control 
  4. Fluoresce quantification of antioxidant enzymes needs to clarify what control was used. Also, comment on the non-linear dose response in CF1553. 
  5. Lipid peroxidation - clarification about the controls. Also, the authors deviated from the 0.1, 0.2, and 0.25 concentrations in this experiment and used 0.5. The concentration of interventions must be kept constant in all the experiments.
  6. The discussion section should focus on explaining and evaluating the findings, showcase how it fits into the current literature, and make an argument supporting the overall conclusion. 

Minor: 

Fig 1. Boxes and circles are hard to differentiate, and the use of colors will improve the readability of the figure. 

The incorporation of colors in all of the figures will significantly improve their readability.

Author Response

In this study, the authors used C. elegans as a model to investigate the in vivo toxicity of an antimicrobial peptide nisin encapsulated in nanoliposomes. Here, using physiological and biochemical experiments, the authors provide evidence for liposome encapsulation-mediated reduced toxicity of nisin in C. elegans.

 Although the authors have performed the required experiments, they still need to include appropriate controls. Also, the methodology, experimental result description, and discussion must be improved. 

Answer: Authors thank the Reviewer for the time and effort to evaluate the manuscript and for recognizing the merits of the study. The manuscript was improved in accordance to the pertinent remarks.

The incorporation of the following suggestions will improve the result interpretations.

  1. Dose-response curve - Vehicle control is required. Also, it should be clearly stated at what time points were chosen and why. A survival plot of C. elegans treated with nisin (0.239 mg mL-1) along with appropriate controls will be helpful.

Answer: Information about vehicle control is included in the figure legend (lines 110-112). The survival of nematodes was determined after incubation for 24 h; this information was described in the Methods section (lines 365-366) and included in the figure legend as well (lines 110-112). In addition, the survival plot of empty liposomes used as control was included.

  1. Development of nematodes - It needs to be clarified from the graph what control was used. It is counter-intuitive that the worms treated with Nisin+liposome 0.1 mg/mL-1 have larger body sizes, and the Authors should comment on it. Typically, developmental assay in C. elegans includes tracking the development of worms from the egg to the adult stage. This can be done by using a molting GFP marker or visually assessing the developmental stage. 

Answer: Controls used were included in the caption to Figure 2 (lines 131-134). Additional discussion on larger body size of nematodes by nisin+liposome 0.1 mg/ml was included (lines 252-257). In this study, the effect of treatments on body size was evaluated after 24 h incubation. No systematic tracking of the development form egg to adult stage performed, but important visual differences were not observed by casual microscopic evaluation.

  1. ROS levels - must include vehicle control

Answer: Information about vehicle control was included in the table footnote (line 149 in the revised manuscript).

  1. Fluoresce quantification of antioxidant enzymes needs to clarify what control was used. Also, comment on the non-linear dose response in CF1553. 

Answer: Indication about controls was included in the figure legend (lines 173-176). Discussion about the non-linear dose response in CF1553 (SOD) was included in the revised manuscript (lines 161-163 and 288-294).

Lipid peroxidation - clarification about the controls. Also, the authors deviated from the 0.1, 0.2, and 0.25 concentrations in this experiment and used 0.5. The concentration of interventions must be kept constant in all the experiments.

Answer: Indication about controls was included in the figure legend (lines 191-194). In this experiment, sufficient amounts were missing for some samples, and for this reason it did not follow the same concentrations. However, as increased TBARS values observed for all treatments, including the lower concentrations, it could be deduced that lipid peroxidation was induced by free and encapsulated nisin, and even by empty liposomes.

The discussion section should focus on explaining and evaluating the findings, showcase how it fits into the current literature, and make an argument supporting the overall conclusion.

Answer: Discussion section was improved as suggested (lines 219-226, 252-254, 288-294 and 307-309).

Minor: 

Fig 1. Boxes and circles are hard to differentiate, and the use of colors will improve the readability of the figure.

Answer: Figure 1 was reformulated using colors to provide a better visualization.

The incorporation of colors in all of the figures will significantly improve their readability.

Answer: Color was incorporated in all figures to provide a better visualization.

Round 2

Reviewer 1 Report

The authors have improved the manuscript. Thus, I would recommend for publication in Molecules.

Author Response

The authors have improved the manuscript. Thus, I would recommend for publication in Molecules.

Answer: Authors thank the Reviewer for the time and effort to evaluate the manuscript and for recognizing the merits of the study.

Reviewer 3 Report

The authors have incorporated suggested changes. However, the authors need to incorporate the following information to support their claims. 

  1. The author's use of letters (a, b, and c) instead of the general "* " to denote p-values is confusing. Authors should use "ns", "*", "**", "***" to denote p-values.
  2. The authors should also clarify, statistical comparison performed with respect to which controls.
  3. Authors should include values used for these comparisons. e.g., mean survival values, body area, etc
  4. Previously for the Lipid peroxidation experimentation, I asked for clarification on the controls. The authors deviated from the 0.1, 0.2, and 0.25 concentrations in this experiment and used 0.5. The concentration of interventions must be kept constant in all the experiments. However, because of limited resources (a common problem in resource constraints developing nations), authors could not complete this assay with previously selected concentrations. This explanation is acceptable because using this selected concentration does not alter the conclusions of this experiment. 

Author Response

The authors have incorporated suggested changes. However, the authors need to incorporate the following information to support their claims.

Answer: Authors thank the Reviewer for the time and effort to evaluate the manuscript and for recognizing the merits of the study. The manuscript was improved in accordance to the pertinent remarks.

1) The author's use of letters (a, b, and c) instead of the general "* " to denote p-values is confusing. Authors should use "ns", "*", "**", "***" to denote p-values.

Answer: The symbol “*” was used to denote p-values as suggested (lines 137-138, 152-153, 181-182, 199 in the revised manuscript).

2) The authors should also clarify, statistical comparison performed with respect to which controls.

Answer: Statistical comparisons were performed comparing the treatments with the control saline solution. This information is provided in the revised manuscript (lines 137-138, 147, 152-153, 181-182, 199).

3) Authors should include values used for these comparisons. e.g., mean survival values, body area, etc

Answer: Additional values used for comparisons were included in the text (lines 101, 119, 122-123, 187-190 in the revised manuscript). Values for DCF fluorescence were directly expressed as percentage of control saline, set as 100% (line 147 in the revised manuscript).

4) Previously for the Lipid peroxidation experimentation, I asked for clarification on the controls. The authors deviated from the 0.1, 0.2, and 0.25 concentrations in this experiment and used 0.5. The concentration of interventions must be kept constant in all the experiments. However, because of limited resources (a common problem in resource constraints developing nations), authors could not complete this assay with previously selected concentrations. This explanation is acceptable because using this selected concentration does not alter the conclusions of this experiment. 

Answer: Authors thank the Reviewer for recognizing the reason why different concentrations were used in the lipid peroxidation experiment.